# Optimization Based on Toughness and Splitting Tensile Strength of Steel-Fiber-Reinforced Concrete Incorporating Silica Fume Using Response Surface Method

**DOI:** 10.3390/ma15186218

**Published:** 2022-09-07

**Authors:** Fuat Köksal, Ahmet Beycioğlu, Magdalena Dobiszewska

**Affiliations:** 1Civil Engineering Department, Yozgat Bozok University, Yozgat 66900, Turkey; 2Department of Civil Engineering, Adana Alparslan Türkes Science and Technology University, Adana 01250, Turkey; 3Faculty of Civil and Environmental Engineering and Architecture, Bydgoszcz University of Science and Technology, Al. Prof. S. Kaliskiego 7, 85-796 Bydgoszcz, Poland

**Keywords:** steel fiber concrete, toughness, optimization, response surface method

## Abstract

The greatest weakness of concrete as a construction material is its brittleness and low fracture energy absorption capacity until failure occurs. In order to improve concrete strength and durability, silica fume SF is introduced into the mixture, which at the same time leads to an increase in the brittleness of concrete. To improve the ductility and toughness of concrete, short steel fibers have been incorporated into concrete. Steel fibers and silica fume are jointly preferred for concrete design in order to obtain concrete with high strength and ductility. It is well-known that silica fume content and fiber properties, such as aspect ratio and volume ratio, directly affect the properties of SFRCs. The mixture design of steel-fiber-reinforced concrete (SFRC) with SF addition is a very important issue in terms of economy and performance. In this study, an experimental design was used to study the toughness and splitting tensile strength of SFRC with the response surface method (RSM). The models established by the RSM were used to optimize the design of SFRC in terms of the usage of optimal silica fume content, and optimal steel fiber volume and aspect ratio. Optimum silica fume content and fiber volume ratio values were determined using the D-optimal design method so that the steel fiber volume ratio was at the minimum and the bending toughness and splitting tensile strength were at the maximum. The amount of silica fume used as a cement replacement, aspect ratio, and volume fraction of steel fiber were chosen as independent variables in the experiment. Experimentally obtained mechanical properties of SFRC such as compression, bending, splitting, modulus of elasticity, toughness, and the toughness index were the dependent variables. A good correlation was observed between the dependent and independent variables included in the model. As a result of the optimization, optimum steel fiber volume was determined as 0.70% and silica fume content was determined as 15% for both aspect ratios.

## 1. Introduction

Concrete is accepted as brittle, and it has a low fracture energy absorption capacity until failure occurs. In order to improve fracture energy or toughness of concrete, short steel fibers have been incorporated into concrete. Incorporation of steel fibers into concrete increases the ductility and toughness of the concrete [1,2,3,4,5,6,7]. A significant effect of steel fibers used in concrete is observed after cracks happen in the matrix. Steel fibers, which are randomly distributed and oriented in the matrix, arrest or stitch the cracks. They present a bridging mechanism transferring the energy. Thus, crack formations are delayed. Furthermore, crack propagation is restrained due to the pull-out process to which steel fibers are subjected [8,9]. High energy is required in order to pull out the fibers from the matrix, resulting in the toughness being increased [10,11]. So, the usage of steel fibers is useful so as to decrease the brittle failure of concrete. There are many parameters regarding fibers, such as the type, aspect ratio, volume, strength, etc. affecting the behavior of SFRC. Other factors that affect the fracture energy, ductility, and toughness of concrete are fiber pull-out resistance, fiber orientation, and strength of the matrix [2,6,12]. SFRCs have been used worldwide in many applications such as paving and flooring, precast elements, hydraulic structures, repairing, linings, etc. [13,14].

The mixture proportioning of concrete is the process of selecting the type and quantity of individual constituents to design concrete. This process is very important to design concrete with appropriate properties for a specific type of application. The prescriptive and performance-based approaches are well-known traditional methods for proportioning concrete mixtures. Prescriptive approaches are step-by-step design methodologies. These methods are based on selecting an appropriate water-to-cementitious material ratio (*w*/*c*), air content, admixture dosage, and both fine and coarse aggregate content to achieve a target compressive strength, slump (for workability), and air content (for freeze–thaw durability). Within these approaches, the designer can proportion an acceptable concrete mixture. Although standard proportioning methods started as arbitrary 1-2-3 cement–sand–aggregate volumetric ratio methods established in the early 1900s, today they have evolved into the absolute volume approach defined by the American Concrete Institute (ACI) [15,16,17].

Performance-based techniques are more flexible when compared to the prescriptive approaches. The design of concrete in these techniques is based on many laboratory trial experiments (defined as the trial-and-error method) [18].

It is known that statistics are used in every field in a beneficial way. From the point of view of civil engineering materials research, the studies carried out in recent years have shown that the use of statistical experimental design in cement-based materials has gained attention in the concrete industry. One of the experimental design methods is the response surface method (RSM).

The RSM has gained increasing attention for its use in the optimization of engineering problems and/or industrial processes. This methodology is a combination of mathematical and statistical techniques that are widely used in the area of concrete preparation optimization [19,20].

Kockal and Ozturan investigated the optimization of properties of lightweight fly ash aggregates for suitability in high-strength lightweight fly ash concrete production using RSM via Design-Expert software. In the optimization shown in this research, the relationships between the sintering parameters (temperature, binder content, and binder type) and the three experimentally obtained responses (specific gravity, water absorption, and crushing strength) were established [21].

Mermerdaş et al., investigated the effect of binder content, curing temperature, and curing time on the compressive strength of lightweight geopolymer mortar. The study was conducted in two stages. In the first stage, the researchers applied experiments with different parameters and obtained 336 data samples. In the second stage, they used the experimental data for optimization through the response surface method. Results show that the experimental verification indicates a good agreement with optimized results [22]. Kaya et al., investigated the mechanical properties of lightweight mortars containing different percentages of additional powder materials using the response surface methodology (RSM). The RSM is highly recommended by researchers to evaluate mechanical properties under high temperatures where concrete includes silica fume and vermiculite [23]. Uche et al., investigated the influence of crumb rubber and calcium carbide residue on the durability properties and heat/temperature resistance of self-compacting concrete (SCC). They used the RSM for experimental design. According to the researchers’ claim, the prediction ability of the RSM can reduce experimental work load, cost, and time [24]. Appana et al., aimed to develop models for predicting the mechanical and shrinkage properties of NaOH-pretreated crumb rubber concrete and conducted multiobjective optimization using the response surface methodology (RSM). They found that the developed RSM models had high R^2^ values ranging from 78.7% to 98% and the optimization produced NaOH and crumb rubber levels of 10% and 2%, respectively, with a high desirability of 71.4% [25]. Xia et al., used the RSM to design the proportions of mixes of white high-strength concrete [26]. Ma et al., used the RSM to analyze the effect of stone powder, pulverized fuel ash, and silica fume contents on the compressive strength of manufactured sand concrete [27].

Awolusi et al. [28] demonstrated the usage of the RSM in relation to the fresh and hardened properties of concrete containing limestone powder reinforced with waste tire steel fiber. The aspect ratio, water/cement ratio, and cement content were established as independent variables while limestone powder was kept constant at 5% by the weight of concrete. The RSM was utilized for predicting water absorption, compressive, flexural, and split tensile strength, and slump of fiber-reinforced concrete. The RSM model predictive efficiency was validated, and a good correlation was observed between the experimental and predicted values. Sinkhonde et al. [29] investigated the compressive strength of rubberized concrete containing burnt clay brick powder using mixes generated by response surface methodology (RSM). The influence of replacement variables of burnt clay brick powder and waste tire rubber on concrete production, cost, and concrete compressive strength responses was assessed. The accuracy of the mathematical models, which were developed using the RSM with findings from experimental design, was tested using analysis of variance (ANOVA). The authors demonstrated that the empirical findings were well-suited to linear and quadratic models for cost and compressive strength responses, respectively. Kumar and Baskar [30] utilized RSM to analyze the fresh and hardened properties of concrete with E-waste plastic (high-impact polystyrene HIPS) as a partial replacement for coarse aggregate. Between the HIPS amount and water/cement ratio, and response variables (slump, density, compressive strength, split tensile strength, and flexural strength) statistical models and final mathematical models in terms of coded factors from predicted responses were developed. Based on the validation of experiments, the authors have shown that the experimental value closely agreed with the predicted value, which validates the calculated response surface models with desirability = 1. The application of response surface methodology in the optimization of fly ash geopolymer concrete was demonstrated by Sun et al. [31]. The authors used a design method of response surface methodology (RSM) to optimize the water/binder ratio, dosage of alkali, unit water dosage, and sodium silicate modulus in relation to compressive strength development. They proved the effectiveness of the RSM in optimizing the preparation of geopolymer concrete through a validation test. Pinheiro et al. [32] applied the RSM to design the experiment required to optimize the composition of an alkaline cement based on ladle furnace slag, fly ash as a precursor, and an alkaline solution prepared with sodium silicate and sodium hydroxide as an activator. The precursor index, activator index, and the sodium hydroxide concentration were considered variables, whereas compression strength and the flexural strength, after 7 and 28 days of curing, were the output variables. The authors demonstrated that the application of the RSM provided the regression equations for the compressive and flexural strength based on the mixture composition. The response surface methods were also used to optimize alkali-activated binders or concrete pastes with different objectives by Rivera et al. [33] and Nunes et al. [34]. The response surface methodology was employed by Vasudevan et al. [35] to optimize the compressive strength, permeability and sorptivity of concrete when two types of admixtures (metakaolin and waste paper sludge ash) were varied simultaneously. Prediction models based on regression analysis from the experimentally obtained data were developed and response contours showing the interaction between the different parameters investigated were established.

In this work, an experimental design was used to study the toughness and splitting tensile strength of SFRC with a silica fume addition. The novelty of this research was to perform a multiobjective simultaneous optimization using the response surface methodology (RSM) in relation to the establishment of mathematical models for flexural toughness (*T_f_*) and splitting tensile strength (*f_st_*). This method was used to optimize both silica fume and steel fiber content to obtain the highest possible flexural toughness (*T_f_*) and splitting tensile strength (*f_st_*) with the lowest possible steel fiber content.

## 2. Response Surface Methodology

The RSM is a combination of mathematics and statistics to build the experimental model. It includes the fitting of regression surface in order to obtain a close-range response. It is also used to design experiments, variations in responses, and designs by the usage of responses approximated. The investigation, which initially generated interest in the package of techniques, was presented by Box and Wilson in 1951. In the earlier works, the RSM was only used to build a model for the experimental responses. Later, the RSM was used for modeling numerical experiments [36,37]. The response surface model can be used to solve inverse problems. It presents three benefits while analyzing the inverse problem. Inverse problems are approximately dealt with without taking into account mechanisms of fracture. In point of many composites, understanding the exact fracture behavior is highly difficult. The second benefit of RSM is the easy evaluation of responses by means of statistical tools. Another benefit is getting less variations in responses found via DOE. Generally, the structure of the relationship between the response and free factors is not known. The initial step in the RSM is obtaining a more fitted estimation for actual correlation. Because of simplicity, functions for approximation are low-order polynomials for many response surfaces, even if functions are unlimited to polynomials. For quadratic polynomials, the response surface is defined in Equation (1):(1)$$y=a_{0}+\sum_{i=1}^{k}a_{i}x_{i}+\sum_{i=1}^{k}a_{ii}x_{i}^{2}+\sum_{i=1}^{k}\sum_{i=1}^{k}a_{ij}x_{i}x_{j}\;\;i<j$$
where *k* is the number of variables. For two variables, the response surface is defined in Equation (2):(2)$$y=a_{0}+a_{1}x_{1}+a_{2}x_{2}+a_{3}x_{1}^{2}+a_{4}x_{2}^{2}+a_{5}x_{1}x_{2}$$

With replacements of $x_{3}=x_{1}^{2}$, $x_{4}=x_{2}^{2}$ and $x_{5}=x_{1}x_{2}$ Equation (2) changes to linear regression in Equation (3):(3)$$y=a_{0}+a_{1}x_{1}+a_{2}x_{2}+a_{3}x_{3}+a_{4}x_{4}+a_{5}x_{5}$$

The total experiment number is *n*. Response surface is defined in Equation (4) and Equation (5):(4)$$\tilde{Y}=\tilde{X}\tilde{A}+\tilde{e}$$
where
(5)$$\tilde{Y}=\left[\begin{array}{l} y_{1}\\ y_{2}\\ .\\ .\\ .\\ y_{n} \end{array}\right]\tilde{X}=\left[\begin{array}{cccccc} 1 & x_{11} & x_{12} & . & . & x_{1k}\\ 1 & x_{21} & x_{22} & . & . & x_{2k}\\ . & . & . & . & . & .\\ . & . & . & . & . & .\\ . & . & . & . & . & .\\ 1 & x_{n1} & x_{n2} & . & . & x_{nk} \end{array}\right]\tilde{A}=\left[\begin{array}{l} a_{0}\\ a_{1}\\ .\\ .\\ .\\ a_{n} \end{array}\right]\tilde{e}=\left[\begin{array}{l} e_{1}\\ e_{2}\\ .\\ .\\ .\\ e_{n} \end{array}\right]$$
where $\tilde{e}$ is the error vector. The impartial predictor $\tilde{b}$ of parameter vector $\tilde{A}$ comes from least square error method in Equation (6):(6)$$\tilde{b}={\left({\tilde{X}}^{T}X\right)}^{-1}{\tilde{X}}^{T}\tilde{Y}$$

Variance–covariance matrix of $\tilde{b}$ comes from Equation (7):(7)$$\mathrm{cov}(b_{i},b_{j})=C_{ij}=s^{2}{\left({\tilde{X}}^{T}\tilde{X}\right)}^{-1}$$
where *s* is the error of $\tilde{Y}$. The predicted score of *s* comes from Equation (8):(8)$$s^{2}=\frac{SS_{E}}{n-k-1}$$

*SS_E_* is the sum of the squares of the residuals, and it is given in Equation (9):(9)$$SS_{E}={\tilde{Y}}^{T}\tilde{Y}-{\tilde{b}}^{T}{\tilde{X}}^{T}\tilde{Y}$$

Understanding the prediction fineness of the response surface, the adjusted coefficient of multiple determination *R*^2^*_adj_* in Equation (10) can be utilized.
(10)$$R_{adj.}^{2}=1-\frac{SS_{E}/(n-k-1)}{S_{yy}/(n-1)}$$
where *S**yy* given in Equation (11) stands for sum of squares in total.
(11)$$S_{yy}^{}={\tilde{Y}}^{T}\tilde{Y}-\frac{(\sum_{i=1}^{n}y_{i}{)}^{2}}{n}$$

The *R*^2^*_adj_* can have 0 or 1, which is the minimum or maximum. Getting closer to 1 means that the response surface is perfect. Results obtained from the response surface are subjected to the t-statistic of the coefficient, bj, in Equation (12):(12)$$t_{0}=\frac{b_{j}}{\sqrt{s^{2}C_{jj}}}$$
where *C**jj* is the element of number *jj* from Equation (7). If the absolute value of the $t_{0}$ meets $\left|t_{0}\right|>t_{\alpha /2,n-k-1}$, the null hypothesis $\beta_{j}=0$ is rejected. Here, α is defined as the acceptable probability that the Type I Error will occur or acceptable probability for rejecting the null hypothesis while it is true. β is the probability that the Type II Error will occur or the part of the optional dispersion which is in the nonrejection part of the risky rate. In Equation (12), the *j*th term is judged to be significant for regression. In order to find the best regression, methods such as stepwise and decreasing methods may be used [37].

## 3. Experimental Works

### 3.1. Materials and Methods

Water–cement was 0.38 in the fabrication of concretes. CEM I 42.5R as cement was utilized. Silica fume came from a ferrochrome factory in Antalya, Turkey. Limestone-based aggregates were used. Fine aggregates up to 0–4 mm in size and coarse aggregates up to 4–12 mm and 12–19 mm in size were used. In control concrete, which was a reference concrete, steel fibers and silica fume were not used. The slump value aimed for was 120 ± 20 mm. Water-reducing admixture was used at the rate of 1% by cement amount to see changes in workability. The aspect ratios of steel fibers were 65 and 80. The fiber volumes (*V_f_*) were 0.5% and 1%. SF contents (*V_SF_*) were 0, 5, 10 and 15% as the percentages of cement weight. The mixture compositions are presented in Table 1.

In concrete production, water including the chemical admixture was added after dry mixing of cement, SF, and aggregates. Next, steel fibers were incorporated into the mixture and mixed again. After casting, molds were vibrated for compaction. Molds were demolded after 24 h, and 28 days of standard water curing was applied. The compressive strength test was carried out on 6 cube specimens with a size of 150 × 150 × 150 mm according to ASTM C109 [38]. An elasticity modulus test was performed on 3 cylinders with a size of Ø150 × 300 mm according to ASTM C469 [39]. Furthermore, six disc specimens Ø150 × 60 mm in size were utilized to determine splitting tensile strength according to ASTM C496 [40]. The flexural tensile strength test was carried out according to ASTM C1018 [41] on 3 samples with 150 × 150 × 500 mm. Tests were carried out under 0.5 mm/min loading.

### 3.2. Experimental Design

The multiobjective simultaneous optimization method is used to optimize both SF (*V_SF_*) and steel fiber content (*V_f_*) of SFRC with a special emphasis on toughness (*T_f_*) and splitting tensile strength (*f_st_*). The collection of experimental data is essential in order to build a model or a response surface and to obtain a good agreement between experimental data and the fitted model. After building a model, the optimization is carried out by using the RSM in order to obtain the optimum solution. Without a model, the optimization cannot be carried out. A common method using the response surface to obtain optimum settings is to use two variables (*V_f_* and *V_SF_*) and to use three fractions of *V_f_* (0%, 0.5%, and 1%) with 65 and 80 aspect ratios and to use four ratios of *V_SF_* (0%, 5%, 10%, and 15%). The designs for two independent variables contain 3 × 4 = 12 mixtures for each aspect ratio, as shown in Figure 1.

Twelve experimental data for each response (fracture toughness and splitting tensile strength) of concretes were fitted to the mathematical model by variance analysis (ANOVA). Regression models of mechanical properties of concretes fitted are given below:

Equations (15) and (16) for the mixtures with an aspect ratio of 65:(13)$$f_{st}=3.04+0.23V_{SF}-0.49V_{f}+3.06V_{f}^{2}+0.12V_{SF}V_{f}$$
(14)$$T_{f}=4.32+6.30V_{SF}+202V_{f}-0.32V_{SF}^{2}-60V_{f}^{2}-3.6V_{SF}V_{f}$$

Equations (17) and (18) for the mixtures with an aspect ratio of 80:(15)$$f_{st}=3.07+0.23V_{SF}-0.14V_{f}+3.30V_{f}^{2}+0.046V_{SF}V_{f}$$
(16)$$T_{f}=-9.02+10.45V_{SF}+411.7V_{f}-0.53V_{SF}^{2}-40V_{f}^{2}-12.56V_{SF}V_{f}$$

Response surfaces of *f_st_* and *T* as a function of volume fractions of *SF* and steel fibers with 65 and 80 aspect ratios are given in Figure 2 and Figure 3.

For an aspect ratio of 65, the correlation coefficients (R^2^) for the model of splitting tensile strength (*f_st_*) and flexural toughness (*T_f_*) were found to be 0.90 and 0.94, respectively. Similarly, the correlation coefficients for the aspect ratio of 80 were 0.94 and 0.98, respectively. Correlation coefficients for each model obtained were quite high. The equations presented here can contribute to pre-experimental mix design by predicting the split tensile strengths and flexural toughness of SFRCs in the range of 0–1% steel fiber volume fraction and 0–15% silica fume content in similar studies.

### 3.3. Test Results

Figure 4 and Figure 5 exhibit load–deflection curves. For each aspect ratio of steel fibers at 1%, load–deflection curves presented a close flexural toughness to the average value of flexural toughness of the series. The test results of concretes are presented in Table 2. The area under the curve for the deflection of 10 mm was used to calculate flexural toughness. Changes in the toughness of concretes were determined by the toughness index (TI). The TI is a ratio of the toughness of fiber-reinforced concrete to that of reference concrete [42].

This parameter measures the relative increase in ductility of concrete with fibers to the control. In this study, control concrete was taken as the concrete including neither silica fume nor steel fiber. Connections among splitting tensile strength and SF content at various steel fiber contents for aspect ratios of 65 and 80 are given in Figure 6 and Figure 7 (with error bars), respectively. Similarly, changes in toughness, depending on both silica fume content and steel fiber content, for aspect rates of 65 and 80 are given in Figure 2 and Figure 3, respectively.

### 3.4. Multiobjective Optimization

For optimization of the objective functions, independent variables were changed, meanwhile and separately, a regression model was built. Later, the connections among mixed design variations and responses were calculated. They are expressed in Equations (15)–(18). A numerical design approach, which utilizes the desirability functions (*d_j_*), could be applied to optimize simultaneous responses. The desirability function (*d_j_*) changed between 0 ≤ *d_j_* ≤ 1. The multiobjective design was obtained via a sole composite response (*D*) presented in Equation (19), which is a geometric average of singular desirability functions. *D* was maximized upon a possible region of optimization factors presented in Equation (18), in which *n* is the number of responses used for the design [37,43].
(17)$$D={\left(d_{1}xd_{2}xd_{3}x\ldots \ldots d_{n}\right)}^{\frac{1}{n}}$$
(18)$$\begin{array}{l} 0\%\leq V_{SF}\leq 15\%\\ 0\%\leq V_{f}\leq 1\% \end{array}$$

The highest flexural toughness (*T_f_*) and splitting tensile strength (*f_st_*) are needed in order to obtain a more ductile concrete. The cost of steel fibers is a crucial factor that should be considered in applications. So, the number of fibers should be kept at a minimum quantity for the economical mixture. Numerical optimization can be carried out by applying each integration of either factors or responses. For this, maximizing the *T* and the *f_st_* is very important while minimizing the *V_f_*. Three responses (*T*, *f_st_*, and *V_f_*) were thought to have equal significance and to be optimized. For *n* = 3, Equation (19) has the form:(19)$$D={\left(d_{1}xd_{2}xd_{3}\right)}^{\frac{1}{3}}$$

The multiobjective designs for the aspect rates of 65 and 80 are presented in Table 3. Figure 8 shows the response surface plots of the *D* for the aspect ratios of 65 and 80. Design-Expert™ software [44] was applied for regression analysis and optimization.

## 4. Conclusions

It is well-known that introducing SF into the mixture so as obtain a higher strength increases concrete brittleness. Usage of fibers in concrete presents a composite with a high ductility because the steel fibers increase the energy absorption capacity of the concrete. Steel fibers and silica fume are jointly preferred for concrete design in order to obtain a concrete with high strength and ductility. Volume fractions of silica fume and steel fibers significantly affect the toughness capacity of concrete. Therefore, their volume fractions in the mixture of concrete should be taken into account to obtain a desirable concrete. The response surface model presents a good approach to optimize the SFRCs while taking into account the many optimization factors such as ductility and cost. Experimental optimization via the RSM also presents an examination of SFRC properties while considering the silica fume content and the number of fibers with two aspect ratios. When the results found in this research were evaluated, the following conclusions were obtained from this investigation:The response surface model presents a good approach to optimize the SFRCs while taking into account the many optimization factors such as ductility and cost.Experimental optimization via the RSM also presents an examination of SFRC properties while considering the silica fume content and the number of fibers with two aspect ratios.If silica fume is used in SFRC mixtures, it should definitely be considered as an independent variable in optimum mixture design. The silica fume had a great influence on the load–deflection behavior of the SFRC.Strengths of SFRCs were significantly increased by increasing silica fume content and the slope of the softening part load–deflection curve in flexure was decreased by using steel fiber. Therefore, both steel fiber and silica fume were quite effective in enhancing the toughness by increasing the area under the load–deflection curve.

Observations from this study exhibit that predictions of quadratic polynomial regressions are able to aid to finding optimal mixtures of concrete. When both mechanical properties (*T* and *f_st_*) and cost optimization were involved, ideal results for the following optimization factors were found: fiber amount of 0.71% and a silica fume content of 15% for an aspect ratio of 65 and, similarly, 0.70% and 15% for an aspect ratio of 80.

## Figures and Tables

**Figure 1 materials-15-06218-f001:**
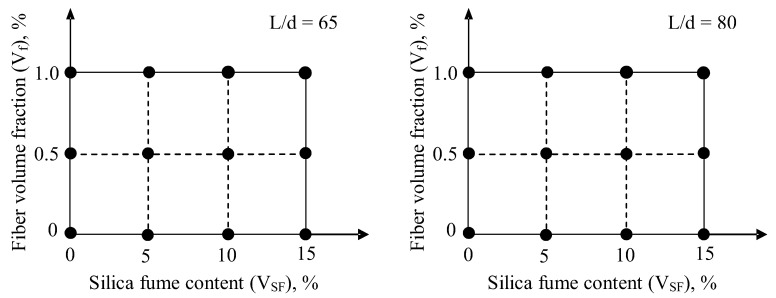
Experimental design schemes.

**Figure 2 materials-15-06218-f002:**
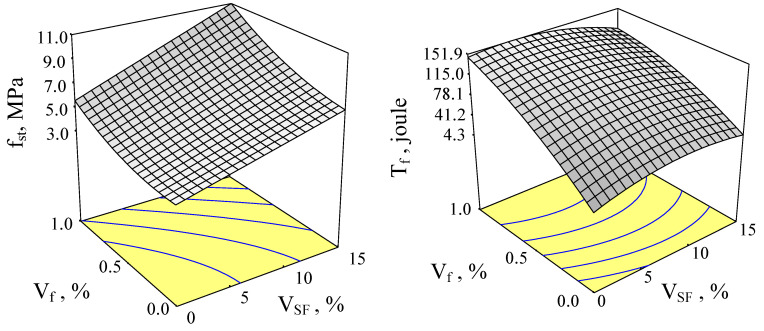
Response surfaces of *f_st_* and *T* for the fiber aspect ratio of 65.

**Figure 3 materials-15-06218-f003:**
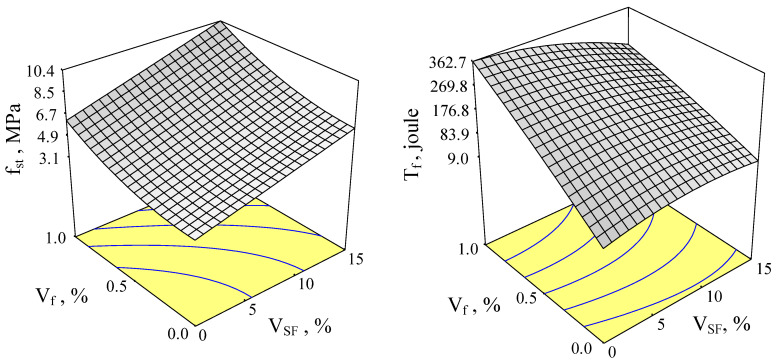
Response surfaces of *f_st_* and *T* for the fiber aspect ratio of 80.

**Figure 4 materials-15-06218-f004:**
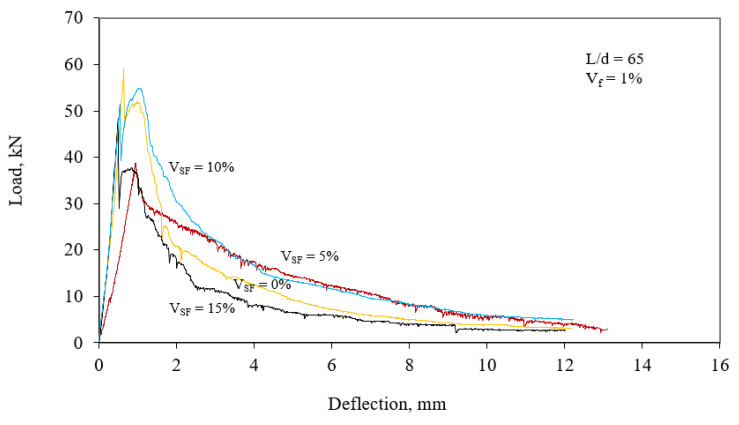
Load–deflection graphs of SFRCs for *V_f_* = 1% and aspect ratio of 65.

**Figure 5 materials-15-06218-f005:**
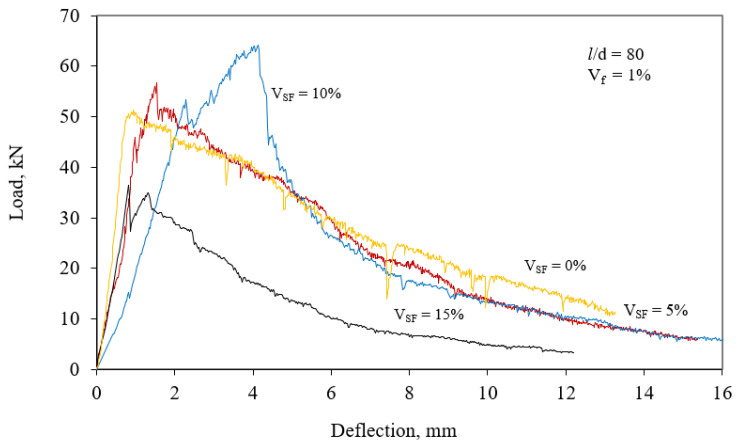
Load–deflection graphs of SFRCs for *V_f_* = 1% and aspect ratio of 80.

**Figure 6 materials-15-06218-f006:**
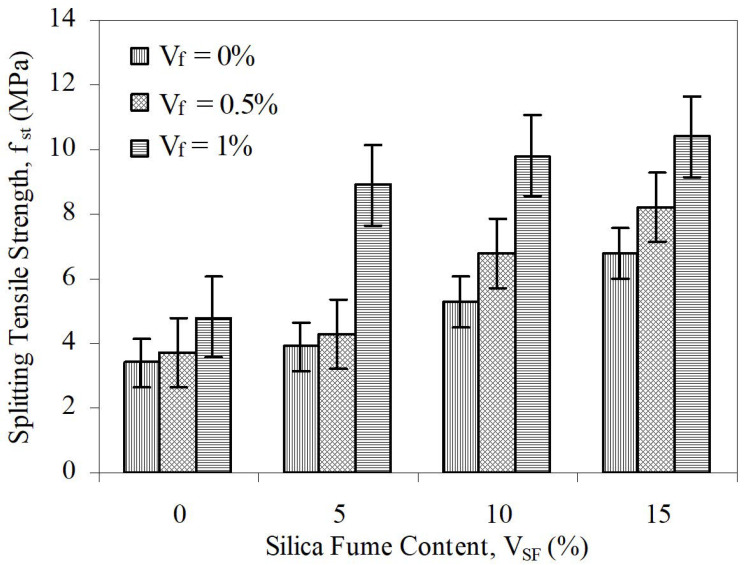
The relation between *f_st_* and *V_SF_* at different *V_f_* for L/d = 65.

**Figure 7 materials-15-06218-f007:**
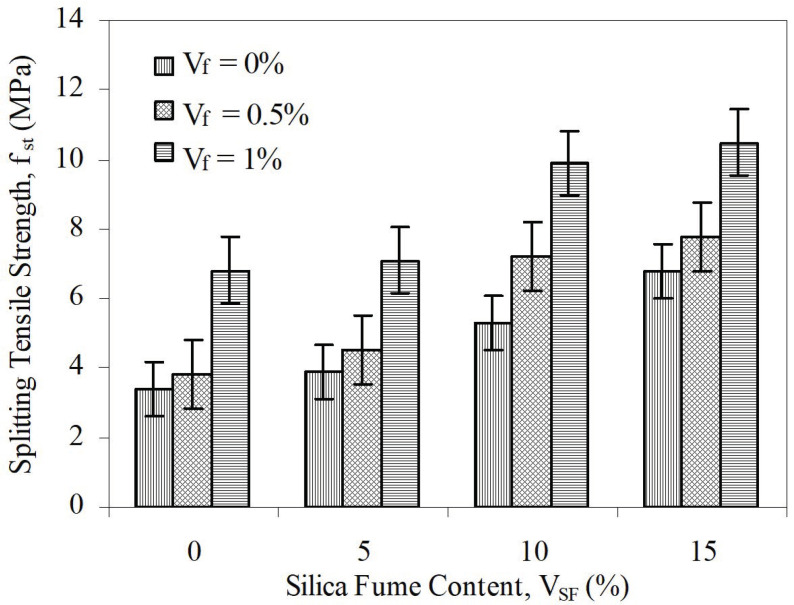
The relation between *f_st_* and *V_SF_* at different *V_f_* for L/d = 80.

**Figure 8 materials-15-06218-f008:**
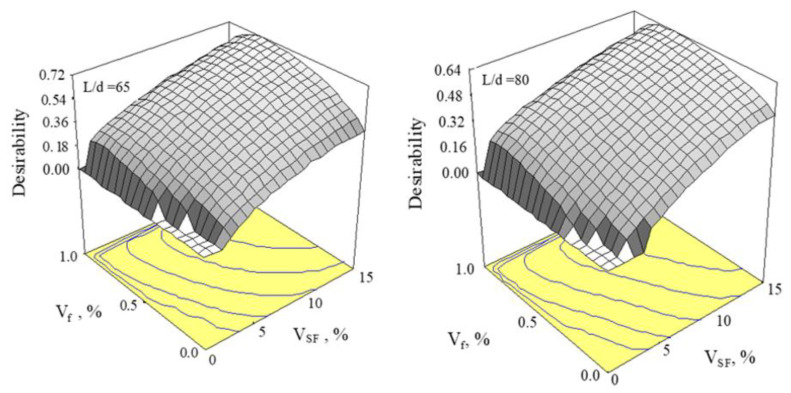
Desirability surfaces for optimal solutions.

**Table 1 materials-15-06218-t001:** Composition of concrete mixtures (kg/m^3^).

Mixture No.	Aspect Ratio	Cement	Water	Water Reducer Admixture	Silica Fume	Steel Fiber	Fine Agg.	Coarse Agg.
1	-	400	152	4	0	0.0	835	1047
2	-	400	152	4	20	0.0	835	1047
3	-	400	152	4	40	0.0	835	1047
4	-	400	152	4	60	0.0	835	1047
5	65	400	152	4	0	39.3	835	1047
6	65	400	152	4	0	78.5	835	1047
7	65	400	152	4	20	39.3	835	1047
8	65	400	152	4	20	78.5	835	1047
9	65	400	152	4	40	39.3	835	1047
10	65	400	152	4	40	78.5	835	1047
11	65	400	152	4	60	39.3	835	1047
12	65	400	152	4	60	78.5	835	1047
13	80	400	152	4	0	39.3	835	1047
14	80	400	152	4	0	78.5	835	1047
15	80	400	152	4	20	39.3	835	1047
16	80	400	152	4	20	78.5	835	1047
17	80	400	152	4	40	39.3	835	1047
18	80	400	152	4	40	78.5	835	1047
19	80	400	152	4	60	39.3	835	1047
20	80	400	152	4	60	78.5	835	1047

**Table 2 materials-15-06218-t002:** Test result of mechanical properties of SFRCs.

Mixture No.	Aspect Ratio (L/d)	Silica Fume (%)	Steel Fiber Content *V_f_*, (%)	Compressive Strength (N/mm^2^)	Elastic Modulus (kN/mm^2^)	Splitting Tensile Strength (N/mm^2^)	Flexural Tensile Strength (N/mm^2^)	Toughness (Joule)	Toughness Index
1	-	0	0	32.4	33.8	3.48	5.7	16	1.0
2	-	5	0	36.4	39.4	3.82	6.1	21	1.3
3	-	10	0	56.2	42.5	5.36	8.1	23	1.4
4	-	15	0	60.1	48.6	6.54	9.4	33	2.1
5	65	0	0.5	33.4	32	3.75	5.9	81	5.1
6	65	0	1.0	37.3	31.6	4.59	6.7	153	9.6
7	65	5	0.5	38.3	37.8	4.05	7.2	95	5.9
8	65	5	1.0	48.1	34.9	8.98	8.7	141	8.8
9	65	10	0.5	60.4	39.1	6.91	8.5	131	8.2
10	65	10	1.0	66.9	38.2	9.56	9.7	152	9.5
11	65	15	0.5	66.5	43.3	8.4	9.5	76	4.8
12	65	15	1.0	69.3	40.6	10.01	10.3	107	6.7
13	80	0	0.5	34.1	32.6	3.7	6.1	166	10.4
14	80	0	1.0	38.5	31.7	6.6	10.1	356	22.3
15	80	5	0.5	41.4	38.1	4.4	7.6	190	11.9
16	80	5	1.0	45.7	35.9	6.9	10.3	360	22.5
17	80	10	0.5	59.7	39.6	7.3	9.0	173	10.8
18	80	10	1.0	63.7	38.4	9.7	11.3	304	19.0
19	80	15	0.5	63.2	46.4	7.5	9.6	159	9.9
20	80	15	1.0	70.5	41.3	10	12.8	183	11.4

**Table 3 materials-15-06218-t003:** Optimum values for silica fume content and steel fiber volume fraction.

Aspect Ratio (L/d)	Silica Fume Content *V_SF_* (%)	Fiber Volume Fraction, *V_f_* (%)	Splitting Tensile Strength, *f_st_* (MPa)	Toughness (Joule)
65	15	0.71	9.0	100.8
80	15	0.70	8.6	166.7

## Data Availability

Not applicable.

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
