# Peer review of "Optimization Based on Toughness and Splitting Tensile Strength of Steel-Fiber-Reinforced Concrete Incorporating Silica Fume Using Response Surface Method"

_materials, 2022, doi:10.3390/ma15186218_

Round 1

Reviewer 1 Report

This article investigates the toughness-based optimization of steel fiber reinforced concrete using the response surface method. There are many fundamental issues involved in the article, requiring some revisions before it can be accepted for publication.

1. Abstract: I suggest starting with placing the question addressed in a broad context and highlighting the study's purpose. Please add one or two lines.

2. Quantify the results and add key findings in the abstract. Please check this statement in the abstract “steel fiber volume ratio is minimum and the bending toughness and splitting tensile strength are maximum”.

3. Please add a few more literature about the RSM to strengthen the literature section.

4. Mention the novelty of this research at the end of the introduction.

5. Define all notations used in equations 1-12.

6. Materials and methods: there is no description of the fracture toughness. What is the method adopted to evaluate fracture toughness? Flexural toughness and fracture toughness are the two terms used in this article. The concept of fracture toughness is different from flexural toughness.

7. What is the reason for selecting the two-aspect ratio of fibres? Furthermore, why is it not considered an independent variable? Are different equations required for the different aspect ratios of fibres?

8. The strength of concrete is based on the cement. Why is cement not considered an independent variable?, since it contributes greatly?

9. Compressive strength can be compared with any parameters and why are the equations predicted for compressive strength?

10. Justify the accuracy of the predicted equations?

11. The equations are developed from the experimental data. Are the equations applicable for global data?

12. Mention the limitations of the equations.

13. “Area under curve for the deflection of 10 mm was used to calculate flexural toughness”. Why is the deflection limited to 10 mm for calculating flexural toughness?

14. The earlier studies should support results.

Author Response

Responses to Reviewers

Reviewer #1

This article investigates the toughness-based optimization of steel fiber reinforced concrete using the response surface method. There are many fundamental issues involved in the article, requiring some revisions before it can be accepted for publication.

  1. Abstract: I suggest starting with placing the question addressed in a broad context and highlighting the study's purpose. Please add one or two lines.

Response to comment 1

Following sentence was added to abstract as first sentence:

“Mixture design of steel fiber reinforced concrete (SFRC) is a very important issue in terms of economy and performance”

  1. Quantify the results and add key findings in the abstract. Please check this statement in the abstract “steel fiber volume ratio is minimum and the bending toughness and splitting tensile strength are maximum”.

Response to comment 2

Following sentence was added to abstract as key findings:

As a result of the optimization, optimum steel fiber volume was determined as 0.70% and silica fume content was determined as 15% for both aspect ratios.

Following last sentence of abstract was removed:

In this study, optimum solutions of the tested SFRC mixtures are given for each aspect ratio of the fiber used.

  1. Please add a few more literature about the RSM to strengthen the literature section.

Response to comment 3

Eight new articles have been analysed and added to the references.

  1. Mention the novelty of this research at the end of the introduction.

Response to comment 4

In this work, an experimental design was aimed for the toughness and splitting tensile strength of SFRC with silica fume addition. The novelty of this research is to perform a multi-objective simultaneous optimalization using response surface methodology (RSM) in relation to establishment mathematical models for flexural toughness (Tf) and splitting tensile strength (fst). This method was used to optimize both, silica fume and steel fibre content to obtain the highest possible flexural toughness (Tf) and splitting tensile strength (fst) with the lowest possible steel fibre content.

  1. Define all notations used in equations 1-12.

Response to comment 5

All notations are explained in the article.

  1. Materials and methods: there is no description of the fracture toughness. What is the method adopted to evaluate fracture toughness? Flexural toughness and fracture toughness are the two terms used in this article. The concept of fracture toughness is different from flexural toughness.

Response to comment 6

Certainly right and it was written wrong. Fracture toughness description was change by flexural toughness in the following sentence:

New one:

“Twelve experimental data for each response (flexural toughness and splitting tensile strength) of concretes, were fitted to the mathematical model by variance analysis (ANOVA).”

Old one and cancelled:

Twelve experimental data for each response (fracture toughness and splitting tensile strength) of concretes, were fitted to the mathematical model by variance analysis (ANOVA).

  1. What is the reason for selecting the two-aspect ratio of fibres? Furthermore, why is it not considered an independent variable? Are different equations required for the different aspect ratios of fibres?

Response to comment 7

Steel fibers with the most commonly used aspect ratios in the SFRC were preferred in the experiment. Of course, aspect ratio could have been an independent variable in the RSM model. If an aspect ratio value was obtained as a result of optimization, which is not available in the market or has no production, then the empirical formulas obtained would not make sense and would not be useful. Therefore, different model was established to obtain different equations for each aspect ratio.

  1. The strength of concrete is based on the cement. Why is cement not considered an independent variable? since it contributes greatly?

Response to comment 8

Of course, cement dosage (or water/cement ratio) is also very important factor for strength of concrete. In this study, cement is no taken as an independent variable because of using silica fume. Therefore, one water/cement ratio was chosen for experimental study.

  1. Compressive strength can be compared with any parameters and why are the equations predicted for compressive strength?

Response to comment 9

As it is well-known, steel fibers improve the tensile property and toughness of concrete while silica fume increases the strength of concrete. Therefore, equations for splitting tensile strength and toughness were derived, taking into account the most significant effects of steel wire and silica fume on concrete.

  1. Justify the accuracy of the predicted equations?

Response to comment 10

Following sentence was added to “3.2. Experimental Design” section:

“For aspect ratio of 65, the correlation coefficients (R2) for the model of splitting tensile strength (fst) and flexural toughness (Tf) were obtained as 0.90 and 0.94, respectively. Similarly, the correlation coefficients for the aspect ratio of 80 were 0.94 and 0.98, respectively.”

  1. The equations are developed from the experimental data. Are the equations applicable for global data?

Response to comment 11

The empirical equations obtained in this study can be used to theoretically predict the splitting and flexural toughness values ​​of concretes containing steel fiber and silica fume with similar properties, before starting to similar investigations to aid in mix design.

  1. Mention the limitations of the equations.

Response to comment 12

Following sentence was added to “3.2. Experimental Design” section:

“Correlation coefficients for each model were obtained quite high. The equations pre-sented here can contribute to pre-experimental mix design by predicting the split ten-sile strengths and flexural toughness of SFRCs in the range of %0-%1 steel fiber vol-ume fraction and %0-%15% silica fume content in similar studies.”

  1. “Area under curve for the deflection of 10 mm was used to calculate flexural toughness”. Why is the deflection limited to 10 mm for calculating flexural toughness?

Response to comment 13

In general, deflection values up to 5, 10 and 15 deflection corresponding to the first crack in the toughness indexes are analysed. In this study, since the initial crack deflection is approximately 1 mm, the flexural toughness was determined by choosing 10 times the initial crack deflection.

  1. The earlier studies should support results.

Response to comment 14

In the Introduction section, examples of successful use of RSM in civil engineering and concrete technology are given. There is no study that is the same or similar to this study and that directly supports the results of this study. This situation also represents the innovative aspect of the article.

Authors would like to thank the Reviewer for all very valuable comments.

Reviewer 2 Report

This study is helpful for the optimization of steel fiber reinforced concrete incorporating silica fume and steel fiber using response surface method. There are still some problems needed to be polished. Some suggestions:

1. The title was not suitable. It only emphasized toughness, steel fiber and Response Surface Methodology. It did not cover the main research elements of this paper.

2. The Abstract was not well organized, especially, the logic was poor. For example, “It is well known that silica fume content and fiber properties such as aspect ratio and volume ratio directly affect the properties of SFRCs. Therefore, an optimization for the mix design of SFRC is required to save time and cost”, it belongs to the research background. The authors are suggested to prepare the Abstract in the order of background, purpose, method, and conclusions. maximum. There is also unclear expression, “Silica fume, aspect ratio and volume fraction of steel fibre were chosen as independent variables…”, What are independent variables chosen from silica fume?

3. The introduction of Response Surface Methodology was unintelligible. Try to explain the method in a clearer way considering the research contents conducted in this paper.

4. In concrete production, the steel fibres were added after water including chemical admixture was added. Did it not cause the agglomeration of steel fibers by doing this?

5. As a general rule, the full name is introduced when the term appears for the first time, and it should be replaced by the abbreviation when it appears again. In the section of Introduction, abbreviation-RMS was directly used, and “response surface methodology (RSM)” appeared twice after it.

6. Please improve some figures, The Figure 6 and Figure 8 were not clearly presented.

7. Please check the writing format of the text. Some sentences are incomplete. “In the study, an experimental design was aimed for the toughness and splitting tensile strength of steel fiber reinforced concrete (SFRC) with the Response Surface Method (RSM)”, please state the aim in detail, such as the aim is to optimize the design of SFRC based on xxx. “Other factors are pull-out resistance, fibre orientation and strength of matrix [6,2,12]”, what exactly do these factors affect? The object should be clear. Misuse of uppercase and lowercase letters, such as “…when compared to the prescriptive approaches. the design of concrete in the techniques…”. Some numbers need to be superscripted, such as “models have high R2 values ranging from…”. The words “fiber” and “fibre” were mixed. There are also many grammar mistakes in the paper. Try to avoid any similar problems.

8. The conclusions are needed to be stated point by point according to the research content. And there is no need to repeat some well-known conclusions.

Author Response

Responses to Reviewers

Reviewer #2

This study is helpful for the optimization of steel fiber reinforced concrete incorporating silica fume and steel fiber using response surface method. There are still some problems needed to be polished. Some suggestions:

  1. The title was not suitable. It only emphasized toughness, steel fiber and Response Surface Methodology. It did not cover the main research elements of this paper.

Response to comment 1

The title has been changed as given below.

Toughness and Splitting Tensile Strength based optimization of steel fiber reinforced concrete incorporating silica fume using response surface method

  1. The Abstract was not well organized, especially, the logic was poor. For example, “It is well known that silica fume content and fiber properties such as aspect ratio and volume ratio directly affect the properties of SFRCs. Therefore, an optimization for the mix design of SFRC is required to save time and cost”, it belongs to the research background. The authors are suggested to prepare the Abstract in the order of background, purpose, method, and conclusions. maximum. There is also unclear expression, “Silica fume, aspect ratio and volume fraction of steel fibre were chosen as independent variables…”, What are independent variables chosen from silica fume?

Response to comment 2

The greatest weakness of concrete as a construction material is its brittleness and a low fracture energy absorption capacity until the failure occurs. In order to improve concrete strength and durability, silica fume SF is introduced into the mixture, which at the same time leads to an increase in the brittleness of concrete. To improve ductile and toughness of concrete, short steel fibres have been incorporated into concrete. Steel fibres and silica fume are jointly preferred for concrete design in order to obtain concrete with high strength and ductile. It is well known that silica fume content and fibre properties such as aspect ratio and volume ratio directly affect the properties of SFRCs. Mixture design of steel fibre reinforced concrete (SFRC) with SF addition is a very important issue in terms of economy and performance. In the study, an experimental design was aimed for the toughness and splitting tensile strength of SFRC with the Response Surface Method (RSM). The models established by RSM are used to obtain the optimum solutions for SFRCs. Optimum silica fume content and fibre volume ratio values ​​were determined using the D-optimal design method so that the steel fibre volume ratio is minimum and the bending toughness and splitting tensile strength are maximum. Amount of silica fume as a cement replacement, aspect ratio and volume fraction of steel fibre were chosen as independent variables in the experiment. Experimentally obtained mechanical properties of SFRC such as compression, bending, splitting, modulus of elasticity, toughness and toughness index were the dependent variables. A good correlation was observed between the dependent and independent variables included in the model. As a result of the optimization, optimum steel fibre volume was determined as 0.70% and silica fume content was determined as 15% for both aspect ratios.

  1. The introduction of Response Surface Methodology was unintelligible. Try to explain the method in a clearer way considering the research contents conducted in this paper.

Response to comment 3

If the reviewer comments that RSM is not being defined enough in the introduction section, we have to mention that since RSM is explained in detail in the "2. Response Surface Methodology" title of the article, it is not given very comprehensively in the introduction section.

  1. In concrete production, the steel fibres were added after water including chemical admixture was added. Did it not cause the agglomeration of steel fibers by doing this?

Response to comment 4

Sometimes steel fibers added to concrete mix before and after water. Generally high steel fiber dosage (> %1) sometimes some agglomeration may be observed. In our study, there was not any agglomeration of steel fibers added after water including chemical admixture during mixture. The fibers were of the type bonded with a water-soluble glue.

  1. As a general rule, the full name is introduced when the term appears for the first time, and it should be replaced by the abbreviation when it appears again. In the section of Introduction, abbreviation-RMS was directly used, and “response surface methodology (RSM)” appeared twice after it.

Response to comment 5

In fact, full name of RSM was firstly introduced in abstract section. Also, following sentence in introduction section was changed as:

“One of the experimental design methods is Response Surface Method RSM.”

  1. Please improve some figures, The Figure 6 and Figure 8 were not clearly presented.

Response to comment 6

The Figure 6 and Figure 8 are changed with clear ones.

  1. Please check the writing format of the text. Some sentences are incomplete. “In the study, an experimental design was aimed for the toughness and splitting tensile strength of steel fiber reinforced concrete (SFRC) with the Response Surface Method (RSM)”, please state the aim in detail, such as the aim is to optimize the design of SFRC based on xxx. “Other factors are pull-out resistance, fibre orientation and strength of matrix [6,2,12]”, what exactly do these factors affect? The object should be clear. Misuse of uppercase and lowercase letters, such as “…when compared to the prescriptive approaches. the design of concrete in the techniques…”. Some numbers need to be superscripted, such as “models have high R2 values ranging from…”. The words “fiber” and “fibre” were mixed. There are also many grammar mistakes in the paper. Try to avoid any similar problems.

Response to comment 7

The models established by RSM are used to optimize the design of SFRC in terms of the usage of optimal silica fume content, and optimal steel fibre volume and aspect ratio.

Other factors affect fracture energy, ductile, and toughness of concrete are fibre pull-out resistance, fibre orientation and strength of matrix [6,2,12].

Grammar mistakes have been corrected.

  1. The conclusions are needed to be stated point by point according to the research content. And there is no need to repeat some well-known conclusions.

Response to comment 8

The conclusions part has been changed according to the reviewer's suggestion. The new version of the conclusions part is as below.

It is well known that the introducing SF into the mixture so as obtain a higher strength increases concrete brittlenes. Usage of fibers in concrete presents composite with a high ductilite because the steel fibers increase the energy absorption capacity of the concrete. Steel fibers and silica fume are jointly preferred for concrete design in order to obtain a concrete with high strength and ductile. Volume fractions of silica fume and steel fibers significantly affect toughness capacity of concrete. Therefore, their volume fractions in the mixture of concrete should be taken account to obtain a concrete which is aimed. Response surface model presents a good approach to optimize the SFRCs while taking account the many optimization factors such as ductility and cost. Experimental Optimization via the RSM also presents an examination of SFRC properties while considering the silica fume content and the amount fibers with two aspect ratios. When the results found in this research are evaluated, the following conclusions can be written from this investigation:

  • Response surface model presents a good approach to optimize the SFRCs while taking account the many optimization factors such as ductility and cost.
  • Experimental Optimization via the RSM also presents an examination of SFRC properties while considering the silica fume content and the amount fibers with two aspect ratios.
  • If silica fume is used in SFRC mixtures, it should definitely be considered as an independent variable in optimum mixture design. So much so that the silica fume had a great influence on the load deflection behavior of the SFRC.
  • Strengths of SFRCs were significantly increased by increasing silica fume content and the slope of the softening part load-deflection curve in flexure was decreased by using steel fiber. Therefore, both steel fiber and silica fume were quite effective on toughness by increasing the area under the load-deflection curve.
  • Observations from this study exhibite that predictions of quadratic polynomial regression are convincing to find optimal mixtures of concrete. When both mechanical properties (T and fst) and cost optimization are involved, ideal results of optimization factors founded are as following: fiber amount of 0.71% and a silica fume content of 15% for aspect ratio of 65 and similarly, 0.70% and 15% for aspect ratio of 80.

Authors would like to thank the Reviewer for all very valuable comments.

Round 2

Reviewer 1 Report

Enough revisions are made by authors

Reviewer 2 Report

The manuscript has been well revised.